# Instability of the Homogeneous Distribution of Chemical Waves in the Belousov–Zhabotinsky Reaction

**DOI:** 10.3390/ma14206177

**Published:** 2021-10-18

**Authors:** Nobuhiko J. Suematsu, Satoshi Nakata

**Affiliations:** 1School of Interdisciplinary Mathematical Sciences, Meiji University, 4-21-1 Nakano, Nakano-ku, Tokyo 164-8525, Japan; 2Graduate School of Advanced Mathematical Sciences, Meiji University, 4-21-1 Nakano, Nakano-ku, Tokyo 164-8525, Japan; 3Meiji Institute for Advanced Study of Mathematical Sciences (MIMS), Meiji University, 4-21-1 Nakano, Nakano-ku, Tokyo 164-8525, Japan; 4Graduate School of Integrated Sciences for Life, Hiroshima University, 1-3-1 Kagamiyama, Higashi-Hiroshima 739-8526, Japan; nakatas@hiroshima-u.ac.jp

**Keywords:** Belousov–Zhabotinsky reaction, wave train, spatiotemporal pattern

## Abstract

Chemical traveling waves play an important role in biological functions, such as the propagation of action potential and signal transduction in the nervous system. Such chemical waves are also observed in inanimate systems and are used to clarify their fundamental properties. In this study, chemical waves were generated with equivalent spacing on an excitable medium of the Belousov–Zhabotinsky reaction. The homogeneous distribution of the waves was unstable and low- and high-density regions were observed. In order to understand the fundamental mechanism of the observations, numerical calculations were performed using a mathematical model, the modified Oregonator model, including photosensitive terms. However, the homogeneous distribution of the traveling waves was stable over time in the numerical results. These results indicate that further modification of the model is required to reproduce our experimental observations and to discover the fundamental mechanism for the destabilization of the homogeneous-distributed chemical traveling waves.

## 1. Introduction

Traveling waves are widely observed in biological systems, such as the action potential propagation on cardiac muscles, which results in the pump function of the heart [1,2,3]. These traveling waves were observed in nonliving chemical systems, such as the Belousov–Zhabotinsky (BZ) reaction, which is a well-known nonlinear chemical reaction that realizes periodic oscillation and ordered pattern formation [4,5]. The fundamental mechanism has been clarified by experimental observations and theoretical approaches using mathematical models, such as the Oregonator [6,7]. These investigations successfully elucidated a variety of phenomena, including the origin of spiral patterns [8,9,10], diode behavior of traveling waves [11,12,13,14], and bifurcation between global oscillation and traveling wave propagation [15,16,17].

The speed of chemical traveling waves depends on environmental conditions, such as the well-known “superspiral pattern” [18,19,20]. If the core of the spiral wave is periodically perturbed by electrical stimuli, the spacing of the chemical waves oscillates over time. As a result, short- and long-spacing regions propagate among the waves, and the long-spacing region forms a spiral, called “superspiral.” Another example is that the speed of chemical waves depends on the period of the spiral core, which depends on chemical conditions [21,22]. A long period of the spiral core generates chemical waves with long spacings, which travel fast. This relationship between the spacing and traveling speed of chemical waves is called the “dispersion relation” [21,22].

The dispersion relation reveals that if there are different spacings, a chemical wave with long spacing closes the gap with the wave in front with a short spacing. One example is the initial inhomogeneous distribution of multiarmed spiral waves approach to homogeneous distribution [23]. Lázár et al. preliminarily demonstrated the behavior of inhomogeneous-distributed wave trains [24]. They prepared a circular excitable medium of the BZ reaction, and unidirectional traveling waves were generated on the medium. Five chemical waves were initially distributed at inhomogeneous distances. The distribution changed over time, and the wave train finally reached a homogeneous distribution [24]. Subsequently, the stability of the homogeneous distribution was systematically investigated using the photosensitive BZ reaction with a change in the number of waves [25], where a circular excitable medium was prepared by controlling the strength of the light illumination, and unidirectional chemical waves were generated at a local position, in which the number of waves changed from 5 to 13. Although all experiments finally reached a homogeneous distribution, the relaxation process depended on the number of waves. The distance between waves monotonically changed for a small number of waves up to eight. However, the spacing between waves oscillated for a large number of waves. In the previous studies, chemical waves were generated in the oscillatory region and led to a circular excitable region [24,25]. Here, a part of the circular excitable region was temporarily cut to eliminate chemical waves traveling on one side, and the circular region was recovered after the desired number of unidirectional chemical waves was prepared. Therefore, the wave train was generated at a local position, and it was technically difficult to prepare homogeneously distributed chemical waves as the initial condition.

In this study, the stability in the homogeneous distribution of a wave train with a high density was investigated by preparing homogeneously distributed chemical waves as the initial condition. To generate chemical waves with a homogeneous distribution, we used a characteristic condition of the photosensitive BZ reaction, in which both photoexcitation and photoinhibition could be realized under the same chemical conditions with different light fields [26]. Under these conditions, steady light illumination inhibits a chemical wave, while a sudden increase in light intensity excites a chemical wave. Using this characteristic photosensitivity, we successfully prepared a homogeneous wave train as the initial condition and investigated its stability. In addition, to support our experimental observation, we also carried out numerical calculations using a modified Oregonator model for such a characteristic photosensitive BZ reaction [26].

## 2. Materials and Methods

Sodium bromate, malonic acid, sulfuric acid, and sodium bromide were purchased from Fujifilm Wako Chemicals Corporation. Ruthenium-tris (2,2′-bipyridyl) dichloride was obtained from Sigma-Aldrich. The chemicals were used without further purification. 

The Belousov–Zhabotinsky (BZ) solution, composed of sodium bromide (0.52 M), sulfuric acid (0.30 M), malonic acid (0.16 M), sodium bromide (0.01 M), and ruthenium bipyridine as the catalyst (1.7 mM). A filter paper (25 mm in the diameter; 1 μm in the pore size) was soaked in the BZ solution for one minute and placed on a glass plate (76 mm in the width; 52 mm in the depth), which was covered with silicone oil to prevent drying. The state of the BZ reaction was controlled by illumination using a liquid-crystal projector (EB-S04, Epson, Japan). To control the light intensity (LI), grayscale images (gray value, 0–255) were created on a personal computer (PC) and projected onto the BZ filter paper (Figure 1). The value of LI at the reaction position monotonically increased with the gray values from 0 to 200 and reached a plateau (1.0 × 10^5^ lx). The value of LI was fitted using the following equation:(1)$$\mathrm{LI}\;=1.7x^{2}+1.4\times 10^{2}x+4.8\times 10^{3},$$
where *x* is the gray value on the PC. The desired image was prepared and controlled using PowerPoint (Microsoft) on the PC.

The experimental setup is shown in Figure 1. The state of the BZ reaction was oscillatory with dark illumination up to 20 and steady state with bright illumination of over 160 of gray value. Under intermediate conditions, the BZ reaction was an excitable medium. In addition, a sudden increase in the light from 50 to 200 of gray value induced excitation of the BZ reaction.

A circular dark region (1.6 × 10^4^ lx) was prepared with a white background (1.0 × 10^5^ lx). The outer and inner diameters of the circular region were 15 and 11 mm, respectively. Thus, the width of the dark excitable region was 2 mm. To generate unidirectional chemical waves in a homogeneous distribution in the circular region, three different images were projected (Figure 2): (i) Narrow lines (0.5 mm in the width) with high intensity (1.0 × 10^5^ lx), which were homogeneously placed in the circular dark region. (ii) Wide lines (1.5 mm in the width) with high intensity (1.0 × 10^5^ lx), which were inserted directly next to the narrow lines of image (i). (iii) Circular dark image. First, the BZ filter paper was illuminated with high intensity (1.0 × 10^5^ lx), and all excited regions disappeared. Image (i) was projected for 2.5 min. The projection was subsequently changed to image (ii) for 5 s and returned to image (i). This short-time irradiation generated chemical waves, which propagated in both directions. After the projection of image (i) for 4 s, it was changed to image (iii), and the behavior of the wave train was observed for at least 5 min. The narrow lines inhibited the chemical waves propagating in a counterclockwise direction, resulting in a wave train propagating in only a clockwise direction. 

## 3. Results

### 3.1. Generation of Unidirectional Chemical Waves

Unidirectional chemical waves were generated using the characteristics of the photosensitive BZ reaction that is photoinhibition and photoexcitation. No chemical waves were generated with the illumination of image (i). However, the wide-line illumination induced excitation of the BZ reaction (Figure 3a), resulting from photoexcitation. This observation agrees with the previous study, where a sudden increase in the light intensity induces excitation of the BZ reaction under this chemical condition [26]. The excitation generated chemical waves in both directions, and the narrow lines inhibited the chemical wave from traveling only in the counterclockwise direction. Therefore, unidirectional chemical waves were generated at the periodic positions (see Appendix A). The chemical wave induction is shown in the space–time diagram for the two waves (Figure 3a,b).

The preparation of homogeneously distributed chemical waves was sensitive to the timing of changes in the light field, and it required careful tuning of the irradiation period of each image. As described above, the chemical waves propagating in the counterclockwise direction were eliminated by the narrow white line, which was irradiated by image (i) after image (ii) was projected (Figure 3b). To successfully generate unidirectional chemical waves, the irradiation period of image (i) played a crucial role. When the period was extremely short, it did not completely inhibit the chemical waves in a counterclockwise direction, which resulted in spiral waves. Oppositely, when the period was extremely long, chemical waves traveling in a clockwise direction contacted the narrow line in front of the waves, and a part of the waves was inhibited, which also resulted in spiral waves (Figure 4). Thus, the irradiation period of image (i) after the photoexcitation had to be adjusted to the appropriate period. Otherwise, spiral waves were generated. We performed seven experiments and succeeded only three times, generating 16 unidirectional chemical waves. However, from a different perspective, our proposed method can be applied to generate spiral cores at the desired positions.

### 3.2. Stability of Homogeneous Distribution of Chemical Waves

The unidirectional chemical waves were homogeneously distributed in the circular excitable region (Figure 5a) and propagated in a clockwise direction. The number of waves was 16. The initial homogeneous distribution gradually broke with the traveling of the wave train, and high- and low-density regions were generated (Figure 5b, see Appendix A). As shown in Figure 5b, there are only two chemical waves per 10 mm in the low-density region and three waves in the high-density region. It was suggested that the equivalent interval of the wave train was unstable under these conditions. Similar behavior was confirmed three times with good reproducibility.

The traveling speed of each chemical wave oscillated over time, which reflected the inhomogeneous distribution of the chemical waves. Initially, all chemical waves accelerated for 50 s, and several waves maintained the traveling speed at approximately 0.15 mm/s. Others further accelerated up to 0.20 mm/s (Figure 6a). Afterward, the latter started to decelerate at *t* = 120 s, while the former started to accelerate again. Eventually, the traveling speed reversed at *t* = 260 s. Similarly, all the chemical waves repeated this acceleration and deceleration, which depended on the spacing from the wave in front. Namely, the traveling speed decelerated and accelerated in the high- and low-density regions, respectively.

The inhomogeneous distribution of the chemical waves was quantitatively indicated using the spacing between the waves. The time series of the spacing from the wave in front was measured by image analysis, as shown in Figure 6b. The spacing for each wave was indicated by brightness; white and black corresponded to 5.0 and 3.0 mm, respectively, and the waves were assigned a number in a clockwise manner (Figure 6b). Under the initial condition, the spacing was distributed from 3.6 to 4.0 mm, which corresponded with the gray color in Figure 6b. The spacing of wave 7 initially increased; that of wave 6 increased thereafter, and the spacing increased one after the other in order (Figure 6b). In other words, the large spacing (white color) propagated among the wave train backward, in a counterclockwise direction. The maximum spacing reached 4.8 mm, and the minimum was 3.2 mm, indicating that the initial homogenous distribution became unstable and low- and high-density regions were generated.

## 4. Discussion

### 4.1. Numerical Simulation Using a Mathematical Model for Photosensitive BZ Reaction

Photoinhibition and photoexcitation of chemical waves were reproduced using a mathematical model [26]. Here, chemical waves were prepared at equivalent spacings, using the same model and conditions. The mathematical model is the Oregonator modified for the photosensitive BZ reaction.
(2)$$\{\begin{array}{c} \frac{\partial u}{\partial t}=D_{u}\nabla^{2}u+\frac{1}{\varepsilon_{1}}\left[qv-uv+u\left(1-u\right)+p_{2}\phi \right],\\ \frac{\partial v}{\partial t}=D_{v}\nabla^{2}v+\frac{1}{\varepsilon_{2}}\left[-qv-uv+fw+p_{1}\phi \right],\\ \frac{\partial w}{\partial t}=D_{w}\nabla^{2}w+u-w+\left(\frac{p_{1}}{2}+p_{2}\right)\phi , \end{array}\;$$
where valuables u, v, and w were the concentrations of HBrO_2_, Br^−^, and Ru(bpy)_3_^3+^, respectively. The parameters were set to $\varepsilon_{1}=0.061,\varepsilon_{2}=2.04\times 10^{-4},D_{u}=D_{v}=1,D_{w}=0.58,q=1.0\times 10^{-4},f=1.2,p_{1}=0.0515,\;and\;p_{2}=0.964$. These parameters corresponded to [H^+^] = 0.6 M, [malonic acid] = 0.16 M, and [BrO_3_^−^] = 0.52. The parameter ϕ represented light intensity and was set to 0.01, 0.02, and 0.04 for the excitable area, wide lines (photoexcitation region), and narrow lines (photoinhibition region), respectively. The numerical simulation was performed in a one-dimensional space with periodic boundary conditions using the Euler method. The length of the space was 100, which corresponded to 3.5 mm in the real space. The time step was $10^{-4}$, and the spatial mesh size was 0.1. 

For the initial condition, all values were set to 0, and the light field ϕ(x) was set to 0.04 for the narrow lines (*x* = 20–25 and 70–75) and 0.01 for the excitable field (*x* = 0–20, 25–70, and 75–100). Owing to the light field, values of u, v, and w changed over time and reached a steady-state (Figure 7(ai,bi)). Thereafter, the value of ϕ(x) at the photoexcitation regions (*x* = 25–50 and 75–100) changed to 0.02. This sudden increase in the light intensity induced the excitation, and u and w drastically increased in the illuminated regions (Figure 7(aii,bii)), resulting in chemical waves in both directions (Figure 7(aiii,biii)). However, the waves traveling to the left contacted the inhibited regions prepared by the narrow lines (Figure 7(aiii,biii)) and disappeared (Figure 7(aiv,biv)). Finally, unidirectional chemical waves traveling to the right were successfully generated (see Appendix A). This is the result of photoinhibition and photoexcitation. 

The two waves propagated in the one-dimensional space and interacted with each other. The traveling behavior is shown in the space–time diagram of w (Figure 8a). The profile of the traveling wave indicated that the activator concentration, u, was high at the front of the wave, and the inhibitor concentration, v, was high behind the wave (Figure 8b). The inhibitor concentration at the front of the wave might have affected the traveling speed of the wave. The initial traveling speed was fast; it drastically decreased over time and approached 4.5 (Figure 8c). In addition, the spacing between the waves was constant over time (Figure 8d). These results indicate that the homogeneous distribution of the traveling waves was stable under this condition.

### 4.2. Stability of the Homogeneous Distribution of Chemical Waves

The number of waves increased to three, or the length of the space decreased to 50. Under both conditions, the traveling speed of the wave approached to constant soon, and the spacing was equal, indicating that the homogeneous distribution of the chemical waves remained stable. A further increase in the number of waves or a decrease in the length of the space induced the disappearance of the waves. Therefore, our numerical calculation indicated that the homogeneous distribution of the traveling waves was stable using the mathematical model and parameter conditions.

The stability of the homogeneous distribution of traveling waves was numerically investigated using other models, such as the modified FitzHugh–Nagumo [27] and Aliev–Panfilov models [28]. These mathematical investigations demonstrated the possibility of the destabilization of the homogeneous distribution of the traveling waves. These studies implied that the Oregonator model required further modification to reproduce our observations and clarify its mechanism.

Our experimental observations showed that the homogeneous distribution of the chemical waves became inhomogeneous (Figure 5 and Figure 6). These behaviors were observed for a short time (5 min), owing to the batch system. Therefore, it was difficult to conclude that the observations were finitely stable or transient. To conclude, further experiments such as a long-time observation using a flow system or a systematic investigation with changes in the number of waves are required. If we find bifurcation of the stability in a homogeneous distribution of wave train, e.g., stable for a small number and unstable for a large number of waves, it strongly supports our hypothesis.

In addition, to uncover the mechanism of the destabilization of the homogeneous distribution, further mathematical approaches for the photosensitive BZ reaction are required. For example, the Oregonator model used here did not consider the total mass of catalyst, and thus, the nullcline for w was almost linear at the region with a large value of u. The Rovinsky–Zhabotinsky model, which is the mathematical model for other catalysts [29], Ferroin, considers the total mass of the catalyst, and its nullcline for w is nonlinear. The mathematical approach using the modified FitzHugh–Nagumo [27] indicates that such modification is one of the possibilities for reproducing experimental observation. 

## 5. Conclusions

In this study, we demonstrated a new experimental method to prepare unidirectional chemical waves at positions with equivalent spacing. This method used two opposite photosensitive characteristics of the BZ reaction: photoinhibition and photoexcitation. Using this method, a homogeneous distribution of unidirectional chemical waves was generated on a circular excitable field, and the destabilization of the homogeneous distribution of the wave train was observed. A similar method was reproduced in the numerical calculations using a mathematical model of the photosensitive BZ reaction. However, the homogeneous traveling waves were stable using the numerical approach. Therefore, to conclude the stability of the homogeneous distribution of the BZ waves, further investigations in the experiments and numerical calculations are required. 

## Figures and Tables

**Figure 1 materials-14-06177-f001:**
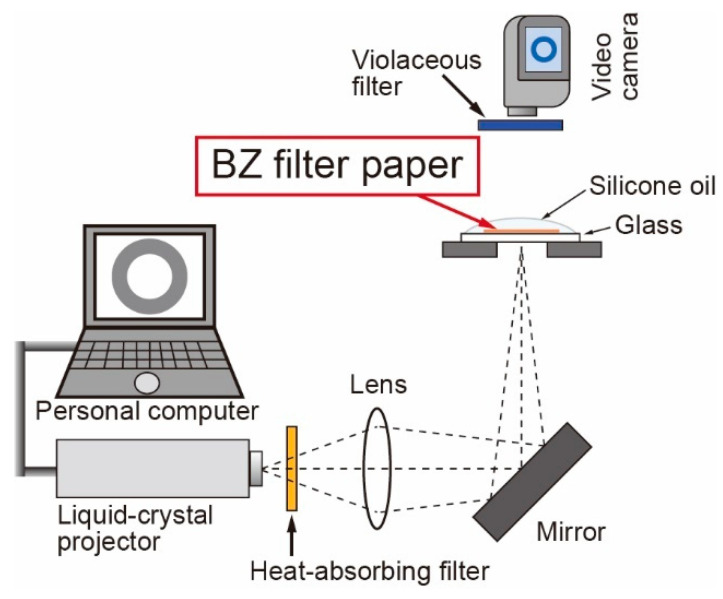
Schematic illustration of the experimental setup. A filter paper was soaked in the Belousov–Zhabotinsky (BZ) solution for one minute and placed on a glass plate, which is indicated as “BZ filter paper”.

**Figure 2 materials-14-06177-f002:**
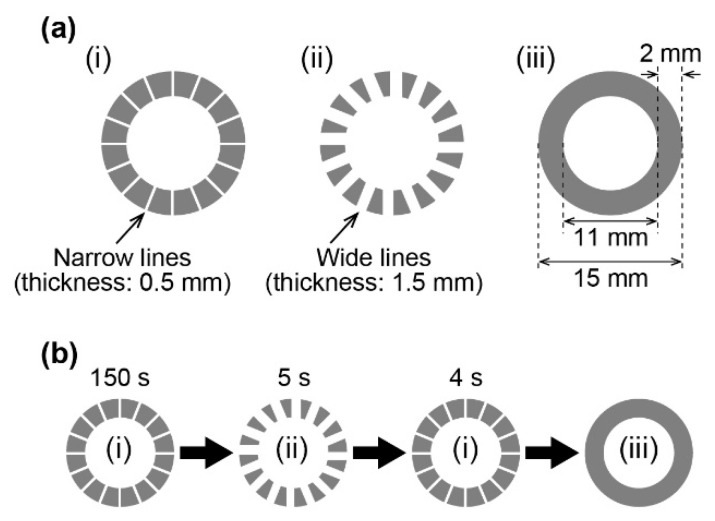
(**a**) Illustration of projected images (top view). (**b**) Time series of change in the projected images.

**Figure 3 materials-14-06177-f003:**
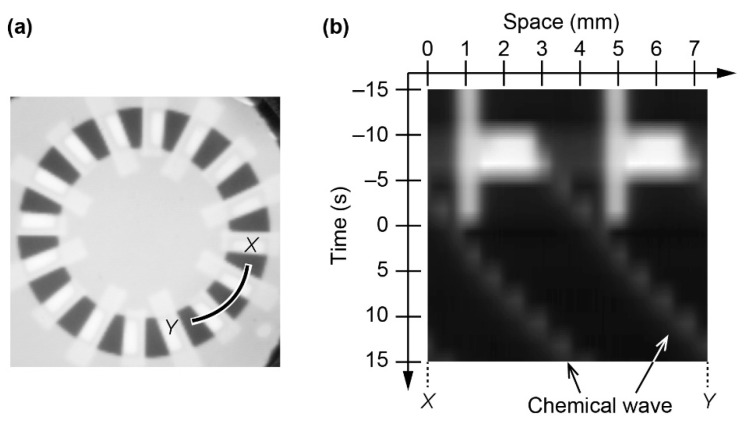
(**a**) Snapshot showing photoexcitation. Rapid increase in the light intensity induced excitation of the BZ reaction. (**b**) Space–time diagram showing the generation of unidirectional chemical waves. The data was obtained on line XY, as indicated in (**a**). The white regions indicate strong illumination, corresponding to narrow and wide lines.

**Figure 4 materials-14-06177-f004:**
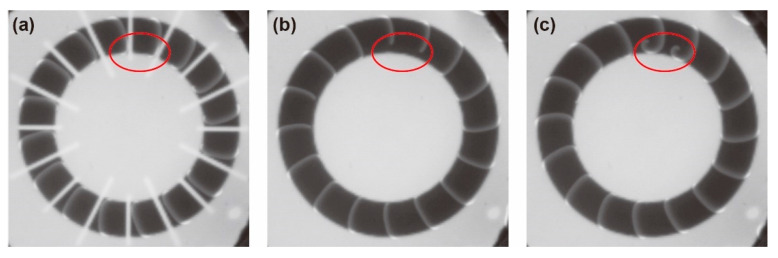
An example of failure to generate unidirectional chemical waves. The snapshots show the process of spiral waves formation owing to the image (i) in Figure 2a irradiated extremely long period. When a part of the chemical waves contacted the narrow line of light illumination (red circle in (**a**)), the contacted regions of the chemical waves were inhibited (**b**). As a result, spiral waves were generated (**c**).

**Figure 5 materials-14-06177-f005:**
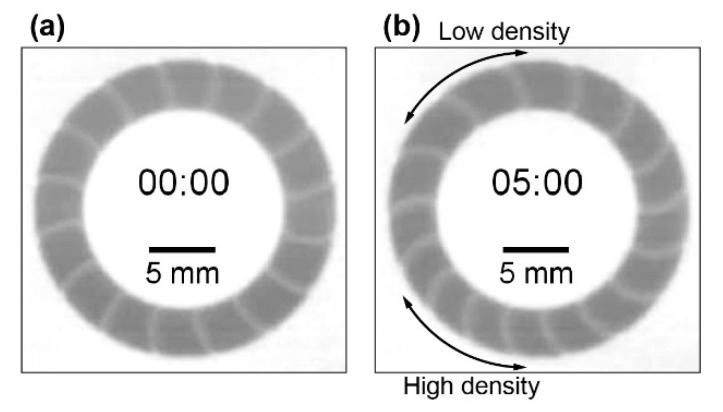
Snapshots at *t* = (**a**) 0 s and (**b**) 300 s. Under the initial condition, 16 chemical waves were distributed with equivalent intervals (homogeneous). Oppositely, low- and high-density regions appeared at *t* = 300 s. The arrows in (**b**) indicate a length of 10 mm. In the low-density region, there were two chemical waves at 10 mm, while three waves were observed in the high-density region.

**Figure 6 materials-14-06177-f006:**
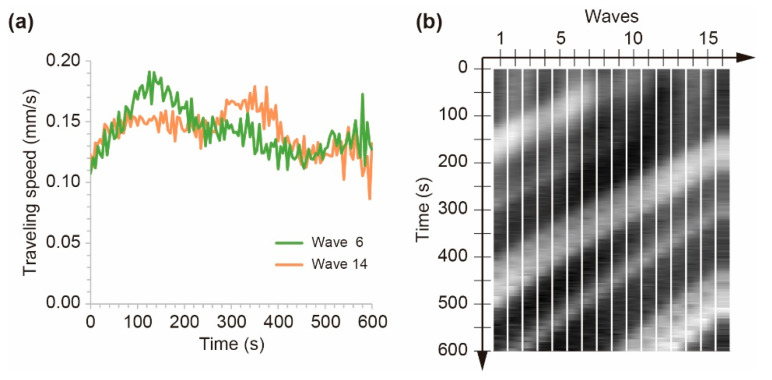
(**a**) Time series of the traveling speed of the chemical waves. (**b**) Time evolution of the spacing from the wave in front. The vertical bars corresponding to each wave. The brightness of each bar indicated the spacing of waves, which were numbered in a clockwise manner from 1 to 16. The width of the bar was appropriately determined and had no meaning.

**Figure 7 materials-14-06177-f007:**
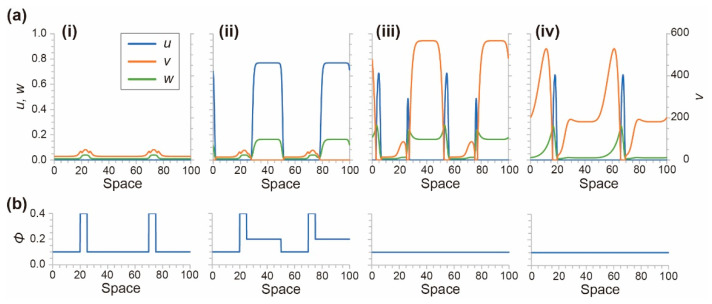
(**a**) Concentration profiles of u, v, and w obtained by numerical calculation. (**b**) The profile of light intensity ϕ(x). (**i**) Steady-state with narrow light illumination within 20–25 and 70–75 (*t* = −2.0). (**ii**) Photoexcitation with wide light illumination within 25–50 and 75–100 (*t* = −0.1). (**iii**) Chemical waves were generated and traveled to the right and left (*t* = 1.0). (**iv**) Chemical waves traveling to the left disappeared because of the narrow light illumination, and only unidirectional waves to the right remained (*t* = 3.0).

**Figure 8 materials-14-06177-f008:**
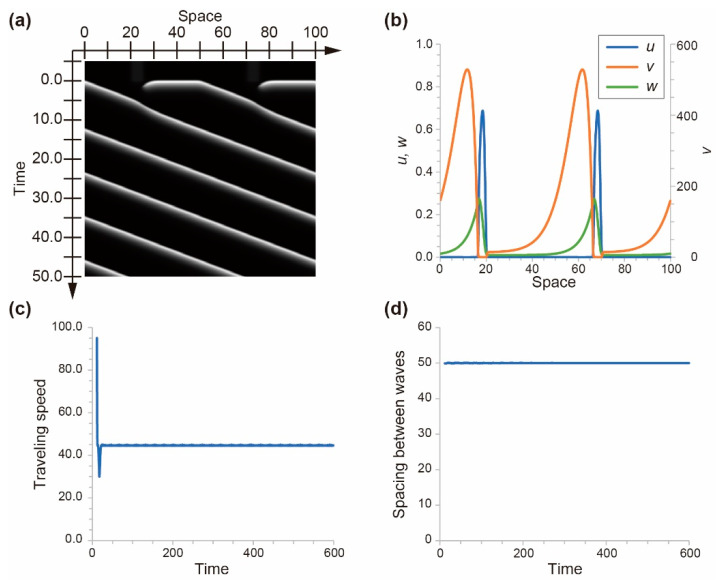
(**a**) Space–time diagram for the value of w. (**b**) Profiles of u, v, and w at *t* = 50.0. (**c**) Time series for the traveling speed of the wave. (**d**) Time series for the spacing between the waves.

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
