# Peer review of "Instability of the Homogeneous Distribution of Chemical Waves in the Belousov–Zhabotinsky Reaction"

_materials, 2021, doi:10.3390/ma14206177_

Round 1

Reviewer 1 Report

Instability of the Homogeneous Distribution of Chemical 
Waves in the Belousov–Zhabotinsky Reaction 
N. J. Suematsu & S Nakata 

Authors have demonstrated throughout the study a new experimental method to prepare unidirectional chemical waves at positions with equivalent spacing by employing two opposite photosensitive characteristics of the BZ reaction: photoinhibition and photoexcitation.  Thus, a homogeneous distribution of unidirectional chemical waves was generated on a circular excitable field, and the destabilization of the homogeneous distribution of  the wave train was observed.

Interestingly, the numerical calculations of a mathematical model of the photosensitive BZ reaction reproduced a similar method. However, the homogeneous traveling waves were stable using the numerical approach.

I think the study is worth publishing once the authors clarify which kind of further studies would be necessary to settle the question, either experimental or numerical. Please take this into account in the discussion of your results.

Author Response

Thank you for your kind comments. As the referee recommended, we added to describe about the future plan to clarify the stability of homogenous distribution of wave train. We have already suggested to change in the number of waves. In order to clearly indicate the plan, we newly added a sentence as below:

291-293: “If we find bifurcation of the stability in homogeneous distribution of wave train, e.g., stable for small number and unstable for large number of waves, it strongly supports our hypothesis.”

Reviewer 2 Report

Authors present interesting results on chemical traveling waves. A new experimental method to prepare chemical waves at positions is demonstrated. This experimental part of the manuscript is very interesting, worth publishing and broad discussion.

Thenumerical calculations were also presented. However, contrary to the experimental results, the homogenous traveling waves were stable using the numerical approach. Some comments about the used model will be valuable. Can, in principle, the Oregonator model (Eq.(2)) reluts to instability (for any parameters).

The paper with small revision can be recommended to publication.

Author Response

Thank you for carefully reading our manuscript. As the referee pointed out, it is worthy to add description about validity of the modified Oregonator model. We added explanation about the Oregonator model modified for photosensitive BZ reaction as follows:

296-301: “For example, the Oregonator model used here did not consider the total mass of catalyst, and thus, the nullcline for w was almost liner at the region with large value of u. The Rovinsky-Zhabotinsky model, which is the mathematical model for other catalyst, Ferroin, considers the total mass of the catalyst [29], and it’s nullcline for w is nonlinear. The mathematical approach using the modified FitzHugh–Nagumo [27] indicates that such modification is one of the possibility for reproducing experimental observation.”

We set the parameters for our numerical calculation based on the experimental condition. So, we have not yet changed those parameters. Thus, it is not sure about the stability of homogenous distribution of wave train using other parameters. However, the parameter region for achieving both photoexcitation and photoinhibition is very narrow, and it was reported that the parameter region to reproduce the both photo responses. So, it is hard to search parameters for checking stability of homogeneous distribution of wave train. In other words, if we change the parameter, it would be hard to prepare traveling waves at the desired positions by our method.

In addition, if we answer to the referee’s question, which is “can the model results to instability in principle?”, we have to mathematically analyze the model. But it is very difficult problem in the field of mathematics. So, we cannot answer the question.

Reviewer 3 Report

The paper presents a numerical investigation on the stability of the homogenous distribution in a wave train by considering the homogenously distributed chemical waves as an initial condition.  Such chemical waves have been generated using a characteristic condition of the photosensitive Belousov‑Zhabotinsky reactions. The work looks interesting. But the presentation does not reflect the actual research work done. Abstract and the last paragraph of Introduction of the paper should be re-written.  

The authors skip to cite the following important reference:

Belousov‑Zhabotinsky type reactions: the non‑linear behavior of chemical systems, Andrea Cassani, Alessandro Monteverde  and Marco Piumetti,  Journal of Mathematical Chemistry (2021) 59:792–826  https://doi.org/10.1007/s10910-021-01223-9

Recommendation: Minor revision.

Author Response

At the beginning, it must note that our manuscript mainly reported experimental results. But the referee considers our manuscript as numerical report. We are not sure why referee took such a big miss understanding.

The referee recommended to re-written the abstract and introduction. But, there is no additional comment why they think so. Therefore, although we are not sure our revision can response the referee’s request, we added following sentence to the end of introduction:

75-77: “In addition, to support our experimental observation, we also carried our numerical calculation using modified Oregonator model for such characteristic photosensitive BZ reaction [26].”

The referee just pointed out to add reference one paper. We carefully read the paper. The paper reported about the BZ reaction, but it includes typical examples and not wave train. We cannot understand why we refer this paper. To introduce the BZ reaction, we have already referred the very important paper written by one of the experts of the nonlinear chemical phenomena including the BZ reaction. We think that there is no reason to additionally refer this paper in our manuscript.